# Implementation of Extended Kalman Filter with Optimized Execution Time for Sensorless Control of a PMSM Using ARM Cortex-M3 Microcontroller

**Justas Dilys** [1,2,*] **, Voitech Stankevič** [1,2] **and Krzysztof Łuksza** [3]

1   State Research Institute Center for Physical Sciences and Technology, Sauletekio Ave. 3,
    LT-10257 Vilnius, Lithuania; voitech.stankevic@ftmc.lt
2   Vilnius Gediminas Technical University, Naugarduko 41, LT-03227 Vilnius, Lithuania
3   Department of Power Electronics and Electrical Machines, Gdansk University of Technology,
    ul. Sobieskiego 7, 80-216 Gdansk, Poland; krzysztof.luksza@pg.edu.pl
*   Correspondence: justas.dilys@ftmc.lt

**Abstract:** This paper addresses the implementation and optimization of an Extended Kalman Filter (EKF) for the Permanent Magnet Synchronous Motor (PMSM) sensorless control using an ARM Cortex-M3 microcontroller. A various optimization levels based on arithmetic calculation reduction was implemented in ARM Cortex-M3 microcontroller. The execution time of EKF estimator was reduced from 260.4 µs to 37.7 µs without loss of accuracy. To further reduce EKF execution time, the separation of a Kalman gain and covariance matrices calculation from prediction and measurement state update, a novel method was proposed, and the performance of it an EKF estimator with separation of a Kalman gain and covariance matrices calculation from prediction and measurement state update was analyzed. Simulation and experiments results validate that the proposed technique could provide the same accuracy with less computation time. A tendency of minimum Kalman gain and covariance matrices calculation frequency from rotor electrical frequency was analyzed and are presented in the paper.

**Keywords:** PMSM; sensorless; EKF; ARM; fast execution

## 1. Introduction

Permanent Magnet Synchronous Motor (PMSM) technology has become attractive thanks to its energy saving capabilities and high dynamic performance. PMSMs have been increasingly used in autonomous electric vehicles, drones, smart buildings and many automation processes [1].

In motor control applications requiring high efficiency, the information about the rotor speed and position is essential to provide feedback for the control loops. The use of mechanical position sensors in motor drives increases the drive's cost and decreases the system's reliability. Therefore, sensorless control would be a practical alternative to the motor control with mechanical sensors. Nowadays, the sensorless control is an essential feature of commercial products in the field of electric motor drives. A popular and widely used sensorless control algorithm is the Extended Kalman Filter (EKF). The EKF is an optimal algorithm which minimizes the mean square error of the estimated quantities. It takes into account the model inaccuracies and measurement noises, and comes up with an accurate estimation result [2,3]. Because of heavy online computation, performed on matrices, the EKF algorithm is a time-consuming process [4]. In order to address this problem, various optimization algorithms which can lower the computational costs have been reported. Computational cost of the full order EKF can be minimized by a reduced order model [5–7]. The idea of these filters is to reduce the number of states of the model by engineering approximation methods. Also, the order reduction simplifies the tuning of the covariance matrices. However, the decrease of the state order can add accuracy damage.

Similarly, the reduced order filter taken with the full model order is obtained by minimizing the trace of the estimation error covariance [8]. However, this method is more practical for systems with large number of states. A novel parallel computational mechanism by defining "useful" data and subdividing computation process is proposed in [9]. In this method, optimization is obtained based on exploiting the numerical characteristics of the system. A various optimization methods for Kalman filter extensions is presented in [10].

Moreover, to provide with system superior robustness and good dynamic performance, online tuning of the electrical parameters is necessary, leaving a small room for the EKF [11]. Therefore, most of the researchers have chosen Digital Signal Processors (DSPs) [12,13] or field programmable logic arrays (FPGAs) [14–17] for the EKF implementation. In a DSP example without simplification, EKF execution time was obtained 71.6 µs [4]. The impressive EKF implementation on DSP with 17 µs execution time of EKF was achieved [18]. While the execution time is short, it is still long for direct torque control, where much shorter sampling period is required compared with field oriented control. In another DSP example, the total execution time of the EKF, matrix converter and all control algorithms fit into 400 µs [19]. While the using FPGA, the EKF execution time 13.36 µs was achieved [20].

However, in many applications the use of DSP processors is not a cost-effective solution. While the cost of the FPGA is lower, the complete system will mostly still require a DSP or another type of processor for the whole system to be implemented. An alternative solution for low-cost and low-power systems are the ARM Cortex-M3 microcontrollers [21]. ARM Cortex-M3 are low-cost, low-power microcontrollers that can replace the existing 8-bit microcontrollers, while still offering 32-bit performance.

In this paper, a strategy is proposed to separate EKF matrices calculation from prediction and measurement update steps, to minimize the overall time consumption of the EKF algorithm. The strategy was simulated with Matlab programming language and implemented on the ARM Cortex-M3 microcontroller.

## 2. State-Space PMSM Model

A dynamic model of a surface-mounted permanent magnet synchronous motor and a sinusoidal flux distribution in a stationary reference frame ($\alpha$, $\beta$) is expressed by the following system of differential equations:

$$
\begin{aligned}
\frac{di_\alpha}{dt} &= -\frac{R_s}{L_s}i_\alpha + \frac{\lambda_m}{L_s}\omega_e \sin\theta_e + \frac{v_\alpha}{L_s} \\
\frac{di_\beta}{dt} &= -\frac{R_s}{L_s}i_\beta - \frac{\lambda_m}{L_s}\omega_e \cos\theta_e + \frac{v_\beta}{L_s} \\
\frac{d\omega_e}{dt} &= \frac{3}{2}\frac{\lambda_m}{J}(i_\beta \cos\theta_e - i_\alpha \sin\theta_e) - \frac{B}{J}\omega_e - \frac{T_L}{J} \\
\frac{d\theta_e}{dt} &= \omega_e
\end{aligned}
\tag{1}
$$

where: $i_\alpha$ and $v_\alpha$ are the $\alpha$ axis current and voltage; $i_\beta$ and $v_\beta$ are the $\beta$ axis current and voltage; $R_s$ is the stator resistance; $L_s$ is the stator phase inductance; P is the number of the pole pairs; $w_e$ and $\theta_e$ are the rotor electrical angular speed and position respectively; $J$ and $B$ are the rotor inertia and viscous damping coefficients respectively; $T_L$ is load the electrical torque. The voltages $v_\alpha$, $v_\beta$ and the load torque $T_L$ are the deterministic control inputs of the system. Both the voltages $v_\alpha$, $v_\beta$ and current $i_\alpha$, $i_\beta$ components are the measurable quantities (2). The stator phase currents $i_a$, $i_b$ and $i_c$ are stator phase currents, which are measured directly.

$$
\begin{aligned}
i_\alpha &= \frac{2}{3}(i_a - \frac{i_b}{2} - \frac{i_c}{2}) \\
i_\beta &= \frac{(i_b - i_c)}{\sqrt{3}}.
\end{aligned}
\tag{2}
$$

The current components in the $\alpha$ - $\beta$ reference frame are obtained from the three phase stator components by a linear transformation [12]. Similar equations hold for the voltages.

## 3. EKF Estimator

The Kalman filter is a mathematical model that runs in parallel to the actual system and provides the estimation of the states of linear systems. It provides a feedback as the difference between the measured output and constantly corrects the model with the error signal. The feedback gain is calculated so that the estimate of state is optimal. The block diagram of Kalman filter are shown in Figure 1.

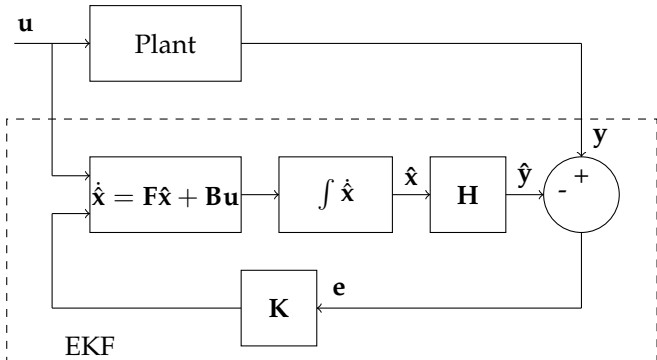

**Figure 1.** Block diagram of Kalman filter.

The state-space model for Kalman filter implementation is derived from (1) with the assumption that the speed $w_e$ is constant during the switching period [22,23].

$$
\begin{aligned}
\frac{di_\alpha}{dt} &= -\frac{R_s}{L_s} i_\alpha + \frac{\lambda_m}{L_s} \omega_e \sin \theta_e + \frac{v_\alpha}{L_s} \\
\frac{di_\beta}{dt} &= -\frac{R_s}{L_s} i_\beta - \frac{\lambda_m}{L_s} \omega_e \cos \theta_e + \frac{v_\beta}{L_s} \\
\frac{d\omega_e}{dt} &= 0 \\
\frac{d\theta_e}{dt} &= \omega_e.
\end{aligned}
\tag{3}
$$

The $\alpha$ - $\beta$ axes stator currents $i_\alpha$, $i_\beta$, the angular electrical rotor speed $w_e$ and position $\theta_e$ are treated as system states. The voltages $v_\alpha$, $v_\beta$ as the input variables. A nonlinear dynamic model accounting for the state transition $\mathbf{w}$ and measurement noise $\mathbf{v}$ can be expressed generally as:

$$
\begin{aligned}
\dot{\mathbf{x}} &= \mathbf{f(x)} + \mathbf{B_c u} + \mathbf{w} \\
\mathbf{y} &= \mathbf{Hx} + \mathbf{v}.
\end{aligned}
\tag{4}
$$

The Gaussian noises $\mathbf{w}$ and $\mathbf{v}$ are white, zero-mean and uncorrelated, and have known covariance matrices $\mathbf{Q}$ and $\mathbf{R}$ respectively. For good EKF performance the choice of the covariance matrices $\mathbf{P}$, $\mathbf{Q}$ and $\mathbf{R}$ is crucial. The covariance matrices $\mathbf{P}$, $\mathbf{Q}$ and $\mathbf{R}$ are symmetric and positive defined symmetric matrices. Covariance matrices give the statistical description of the model inaccuracy. Matrix $\mathbf{Q}$ represents the statistical description of the model, matrix $\mathbf{R}$ indicates the magnitude of measurement noise, matrix $\mathbf{P_0}$ contains the information of variances at the initial conditions and mainly affects the convergence rate of EKF in the transient condition [24]. Since these are usually unknown, in most cases the EKF matrices are designed and tuned by trial-and-error procedures [25].The state vector $\mathbf{x}$ is denoted as $\mathbf{u}$ is input vector and $\mathbf{y}$ is the output vector:

$$
\mathbf{x} = \begin{bmatrix} i_\alpha & i_\beta & \omega_e & \theta_e \end{bmatrix}^T, \quad \mathbf{u} = \begin{bmatrix} V_\alpha & V_\beta \end{bmatrix}^T, \quad \mathbf{y} = \begin{bmatrix} i_\alpha & i_\beta \end{bmatrix}^T.
\tag{5}
$$

The model's matrices are expressed as follows:

$$
\mathbf{f}(\mathbf{x}) = \begin{bmatrix} f_1 \\ f_2 \\ f_3 \\ f_4 \end{bmatrix} = \begin{bmatrix} -\frac{R_s}{L_s} i_\alpha + \frac{\lambda_m}{L_s} \omega_e \sin \theta_e \\ -\frac{R_s}{L_s} i_\beta - \frac{\lambda_m}{L_s} \omega_e \cos \theta_e \\ 0 \\ \omega_e \end{bmatrix},
\tag{6}
$$

$$
\mathbf{B_c} = \begin{bmatrix} \frac{1}{L_s} & 0 \\ 0 & \frac{1}{L_s} \\ 0 & 0 \\ 0 & 0 \end{bmatrix}, \quad \mathbf{H} = \begin{bmatrix} 1 & 0 \\ 0 & 1 \\ 0 & 0 \\ 0 & 0 \end{bmatrix},
\tag{7}
$$

$$
\mathbf{B} = T\mathbf{B_c} = \begin{bmatrix} \frac{T}{L_s} & 0 \\ 0 & \frac{T}{L_s} \\ 0 & 0 \\ 0 & 0 \end{bmatrix}
\tag{8}
$$

where $T$ is sampling time. The PMSM model described by (6) is nonlinear as products of variables are involved. The nonlinear function $\mathbf{f}(\mathbf{x})$ is approximated by a linear set. The continous time Jacobian matrix is expressed as:

$$
\mathbf{F}_c = \left. \frac{\partial \mathbf{f}(\mathbf{x})}{\partial \mathbf{x}} \right|_{\mathbf{x}=\mathbf{x}_{k-1}} = \begin{bmatrix} \frac{\partial \mathbf{f_1}}{\partial i_\alpha} & \frac{\partial \mathbf{f_1}}{\partial i_\beta} & \frac{\partial \mathbf{f_1}}{\partial w_e} & \frac{\partial \mathbf{f_1}}{\partial \theta_e} \\ \frac{\partial \mathbf{f_2}}{\partial i_\alpha} & \frac{\partial \mathbf{f_2}}{\partial i_\beta} & \frac{\partial \mathbf{f_2}}{\partial w_e} & \frac{\partial \mathbf{f_2}}{\partial \theta_e} \\ \frac{\partial \mathbf{f_3}}{\partial i_\alpha} & \frac{\partial \mathbf{f_3}}{\partial i_\beta} & \frac{\partial \mathbf{f_3}}{\partial w_e} & \frac{\partial \mathbf{f_3}}{\partial \theta_e} \\ \frac{\partial \mathbf{f_4}}{\partial i_\alpha} & \frac{\partial \mathbf{f_4}}{\partial i_\beta} & \frac{\partial \mathbf{f_4}}{\partial w_e} & \frac{\partial \mathbf{f_4}}{\partial \theta_e} \end{bmatrix} = \begin{bmatrix} -\frac{R_s}{L_s} & 0 & \frac{\lambda_m}{L_s} \sin \theta_e & \frac{\lambda_m}{L_s} \omega_e \cos \theta_e \\ 0 & -\frac{R_s}{L_s} & -\frac{\lambda_m}{L_s} \cos \theta_e & \frac{\lambda_m}{L_s} \omega_e \sin \theta_e \\ 0 & 0 & 0 & 0 \\ 0 & 0 & 1 & 0 \end{bmatrix}
\tag{9}
$$

where the previous estimate of $\mathbf{x}$ as a reference point is taken for discretization around this point. For digital implementation the system model (4) has to be discretized. The discrete nonlinear dynamic model is expressed as follows:

$$
\dot{\mathbf{x}}_k = \mathbf{F}_{k-1}\mathbf{x}_{k-1} + \mathbf{B}\mathbf{u}_{k-1} + \mathbf{w}_{k-1}
$$
$$
\mathbf{y} = \mathbf{H}\mathbf{x}_k + \mathbf{v}_k
\tag{10}
$$

where in (11) the matrices $\mathbf{x}_k$, $\mathbf{F}_{k-1}$, $\mathbf{u}_{k-1}$ are discrete matrices of $\mathbf{x}$, $\mathbf{F}_c$, $\mathbf{u}$ respectively, and the matrices $\mathbf{w}_{k-1}$, $\mathbf{v}_k$ are discrete matrices of $\mathbf{w}$, $\mathbf{v}$ respectively, independent of the system state. Based on Equation (9) after discretization of $\mathbf{F_c}$ the Jacobian matrix is:

$$
\mathbf{F_k} = \mathbf{F} = e^{\mathbf{F_c}T} \approx \mathbf{I} + \mathbf{F}_c T = \begin{bmatrix} 1 - T\frac{R_s}{L_s} & 0 & T\frac{\lambda_m}{L_s} \sin \theta_e & T\frac{\lambda_m}{L_s} \omega_e \cos \theta_e \\ 0 & 1 - T\frac{R_s}{L_s} & -T\frac{\lambda_m}{L_s} \cos \theta_e & T\frac{\lambda_m}{L_s} \omega_e \sin \theta_e \\ 0 & 0 & 1 & 0 \\ 0 & 0 & T & 1 \end{bmatrix}
\tag{11}
$$

where the matrix $\mathbf{F}$ in (11) is a discrete, linearized Jacobian matrix.

The extended Kalman filter can be realized by the following steps. The first step is a time update of the state vector and the error covariance matrix, in which a prediction based on the previous estimates $\hat{\mathbf{x}}_{k-1}$ is performed:

$$
\mathbf{F}_{k-1} = \mathbf{F}(\mathbf{x} = \hat{\mathbf{x}}_{k-1})
\tag{12}
$$

$$
\hat{\mathbf{x}}_k^- = \hat{\mathbf{x}}_{k-1} + T\mathbf{f}(\hat{\mathbf{x}}_{k-1}) + \mathbf{B}\mathbf{u}_{k-1}
\tag{13}
$$

$$
\mathbf{P}_k^- = \mathbf{P}_{k-1}\mathbf{F}_{k-1}\mathbf{P}_{k-1}{}^T + \mathbf{Q}
\tag{14}
$$

where the Jacobian matrix (12) is computed too. Important to note is that the state prediction is done by integrating Equation (13) with Runge-Kutta or other similar method. The second step is a measurement update that corrects the predicted state estimate $\hat{\mathbf{x}}_k^-$ and its error covariance $\mathbf{P}_k^-$ matrix through a feedback correction:

$$\mathbf{P}_k = (\mathbf{I} - \mathbf{K}_k \mathbf{H})\mathbf{P}_k^- \tag{15}$$

$$\hat{\mathbf{x}}_k = \hat{\mathbf{x}}_k^- + \mathbf{K}_k(\mathbf{y}_k - \mathbf{H}\hat{\mathbf{x}}_k^-) \tag{16}$$

where the extended Kalman filter gain matrix $\mathbf{K}_k$ is:

$$\mathbf{K}_k = \mathbf{P}_k^- \mathbf{H}^T (\mathbf{H}\mathbf{P}_k^- \mathbf{H}^T + \mathbf{R})^{-1} \tag{17}$$

Also the correction of the estimated rotor position (18) to limit the angle to a $2\pi$ interval is added:

$$\hat{\mathbf{x}}_k(4) = \hat{\mathbf{x}}_k(4) - 2\pi k, \quad k = \lfloor \hat{\mathbf{x}}_k(4)/2\pi \rfloor. \tag{18}$$

The estimation error covariance matrix $\mathbf{P}$ denotes the error of the state vector $\hat{x}$ (19).

$$\mathbf{P} = E[(\hat{\mathbf{x}} - \bar{\mathbf{x}})(\hat{\mathbf{x}} - \bar{\mathbf{x}})^T] = \begin{bmatrix} P_{11} & P_{12} & P_{13} & P_{14} \\ P_{21} & P_{22} & P_{23} & P_{24} \\ P_{31} & P_{32} & P_{33} & P_{34} \\ P_{41} & P_{42} & P_{43} & P_{44} \end{bmatrix} \tag{19}$$

where $E[]$ is an operator computing the mean of the variable inside the brackets and $\bar{\mathbf{x}}$ is the mean of the estimated variable. The error covariance matrix $\mathbf{P}$ is a degree of accuracy of the estimate. If $\mathbf{P}$ is large the error of the estimate is large and if is small the error of the estimate is small. The element $P_{44}$ is the variance of the rotor position and it could be an indicator of how well the Kalman filter estimate rotor position. The Kalman gain matrix (20):

$$\mathbf{K} = \begin{bmatrix} K_{11} & K_{12} \\ K_{21} & K_{22} \\ K_{31} & K_{32} \\ K_{41} & K_{42} \end{bmatrix} \tag{20}$$

is used as weighting in the measurement update process. The measurement update Equation (16) corrects the state, accounting for the measurements, and can be expressed as:

$$\begin{aligned} \hat{i}_{\alpha k} &= \hat{i}_{\alpha k}^- + K_{11}(i_{\alpha k} - \hat{i}_{\alpha k}^-) + K_{12}(i_{\beta k} - \hat{i}_{\beta k}^-) \\ \hat{i}_{\beta k} &= \hat{i}_{\beta k}^- + K_{21}(i_{\alpha k} - \hat{i}_{\alpha k}^-) + K_{22}(i_{\beta k} - \hat{i}_{\beta k}^-) \\ \hat{\omega}_{ek} &= \hat{\omega}_{ek}^- + K_{31}(i_{\alpha k} - \hat{i}_{\alpha k}^-) + K_{32}(i_{\beta k} - \hat{i}_{\beta k}^-) \\ \hat{\theta}_{ek} &= \hat{\theta}_{ek}^- + K_{41}(i_{\alpha k} - \hat{i}_{\alpha k}^-) + K_{42}(i_{\beta k} - \hat{i}_{\beta k}^-). \end{aligned} \tag{21}$$

In Equation (21) the elements $K_{41}$ and $K_{42}$ are position correction gains. As only the stator winding currents can be measured, the rotor position and velocity are mainly estimated in measurement update steps.

*EKF Technique with Parallel Calculation*

The EKF calculation is separated into two different procedures. One is the control procedure and the other the background procedure. The sequence of the EKF algorithm implementation is shown in Figure 2 as a flow diagram.

The control procedure is executed on every PWM switching period and predicts the new state vector using (13), updates the predicted state vector using (16). And also corrects the estimated rotor position to a periodic function using (18). The background procedure calculates the Kalman gain and all covariance matrices. The background is executed on every time $m$. The majority of the EKF algorithm computation is performed

in the background procedure and runs in a periodic cycle, just like the control procedure. The calling rate of the background procedure is the same or smaller, compared to the control procedure.

The reduced Kalman gain and all covariance matrices calculation ratio could be treated like PMSM model are constant for period of time equal to background procedure call period. While the Kalman gain and all covariance matrices calculation ratio is reduced the prediction state (13) and measurement correction state (16) are executed on every control cycle. The prediction state (13) is always updated with the newest input variables in every control cycle. The measurement correction state (16) is also fed with the newest measurements, but with the Kalman gains being constant for a period of time equal to control period.

As shown in Figure 3 the background task period $T_b$ is longer or the same compared to $T_s = \frac{1}{F_s}$ where $F_s$ is PWM switching frequency. The background procedure is separated from the control procedure and does not depend on the PWM switching frequency. The switching frequency can be set to a very high value, while the background procedure can run at a much lower frequency. The total process execution time of EKF could be significantly reduced, because the heavy calculation are doing in background task with period longer than control period.

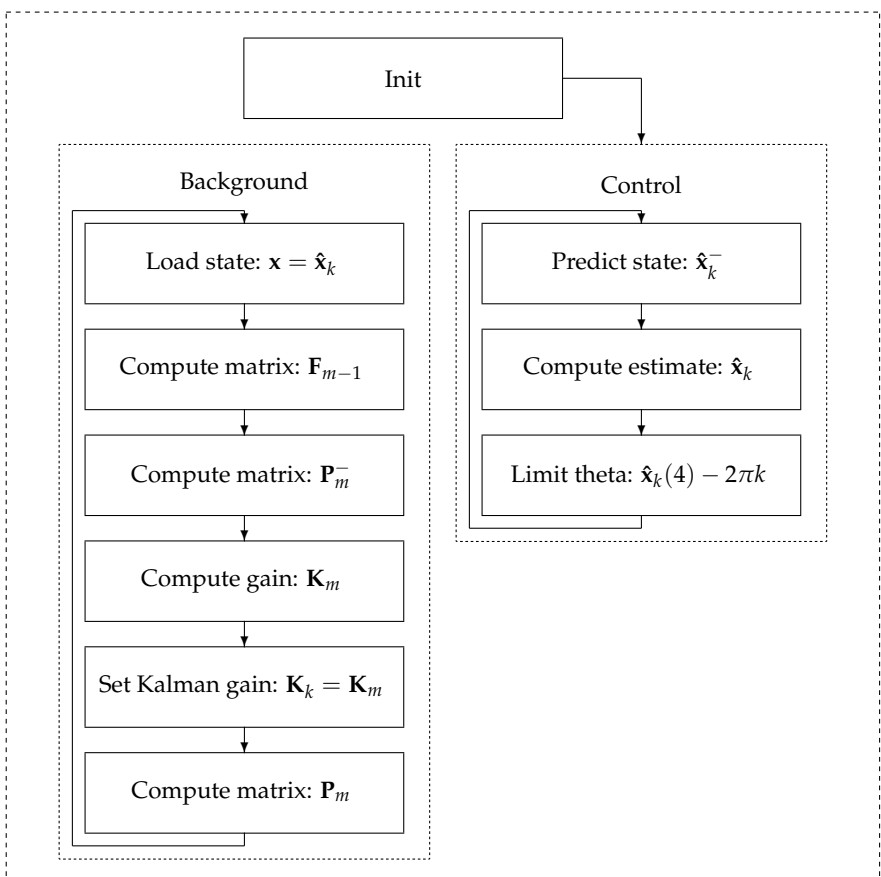

**Figure 2.** Flow diagram ot the EKF algorithm with parallel calculation.

The control procedure has a higher priority and interrupts the background procedure when it is time to execute it. The synchronization is also performed in the background task, the state vector is taken from the control procedure and the Kalman gain is provided to the control procedure after it is computed. After Kalman gain has been computed in the background procedure, it is important immediately to copy the new Kalman gain matrix into the control procedure to allow control procedure more effectively, to use Kalman gain.

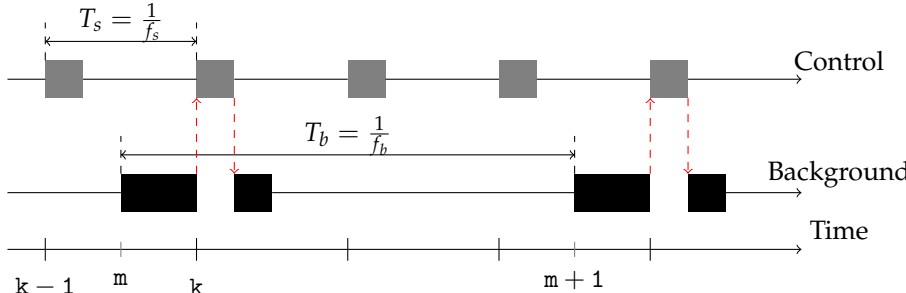

**Figure 3.** Time diagram showing the parallel run of the two EKF algorithm procedures. $T_s$—control period time, $T_b$—background procedure period.

## 4. Simulation Results

Simulations have been performed with the Matlab programming language. To verify the performance of the proposed EKF algorithm, the simulated PMSM parameters were set for the same values as in the experimental setup and are presented in Table 1.

**Table 1.** Simulation parameters.

| | | |
|---|---|---|
| Stator resistance | $Rs$ | 1.2 $\Omega$ |
| Synchronous inductance | $Ld$ | 0.5 mH |
| Synchronous inductance | $Lq$ | 0.5 mH |
| Flux linkage | $\lambda_m$ | 0.007 Wb |
| Number of poles | $P$ | 8 |
| DC supply voltage | $V_{dc}$ | 24 V |
| Switching frequency | $F_s$ | 5 kHz |

In order to make the simulation model consistent with further experimental verification, the space vector pulse width modulation (SVPWM) and overall field oriented control system was simulated. In order to verify the effectiveness of the EKF algorithm based on parallel computation, the simulation was carried out for different set point speed values. The performance of sensorless control for various motor speeds is investigated in simulation. The initial state covariance matrix $\mathbf{P_0}$ and covariance matrices accounting for the model and measurement ($\mathbf{Q}$ and $\mathbf{R}$ respectively) are as follows:

$$\mathbf{P_0} = \begin{bmatrix} 1 & 0 & 0 & 0 \\ 0 & 1 & 0 & 0 \\ 0 & 0 & 1 & 0 \\ 0 & 0 & 0 & 1 \end{bmatrix}, \quad \mathbf{Q} = \begin{bmatrix} 1 & 0 & 0 & 0 \\ 0 & 1 & 0 & 0 \\ 0 & 0 & 500 & 0 \\ 0 & 0 & 0 & 0.1 \end{bmatrix}, \quad \mathbf{R} = \begin{bmatrix} 1 & 0 \\ 0 & 1 \end{bmatrix}. \tag{22}$$

The EKF algorithm in Figure 2 was the main research object. There, the Kalman gain matrix and its covariance matrices calculation is performed in the background procedure while the prediction and update in the control procedure. The different frequency ratios of the background to the control procedure of the EKF performance at various fundamental motor speeds was researched.

The EKF performance, the Kalman gain and all covariance matrices calculated at the same rate of 5 kHz, are shown in Figure 4. In Figure 5 the Kalman gain and all covariance matrices are calculated at a five times slower rate of $f_b = 1$ kHz, while the control procedure execution rate is the same (5 kHz). Comparing Figure 4c with Figure 5c can be seen that the position error did not increase. The position variance $P_{44}$ in Figures 4b and 5b is approximately equal too, only the discretization is different. The Kalman gains $K_{41}$ and $K_{42}$ for position estimation in Figures 4d and 5d have the same amplitude peak values and have similar sine waveform shapes, but the waveform in Figure 5d are more discretized. The discretization is because of reduced calculation frequency of Kalman gain in simulation is taken. From comparison of Figures 4 and 5 we can conclude that the magnitudes of

Kalman gain for position estimation did not change, only it's discretization. For a period of time Kalman gain values are constant values, and they are used multiple time in correction step. Even Kalman gains are constant values the prediction step and correction steps still corrects state vector accounting new measurements.

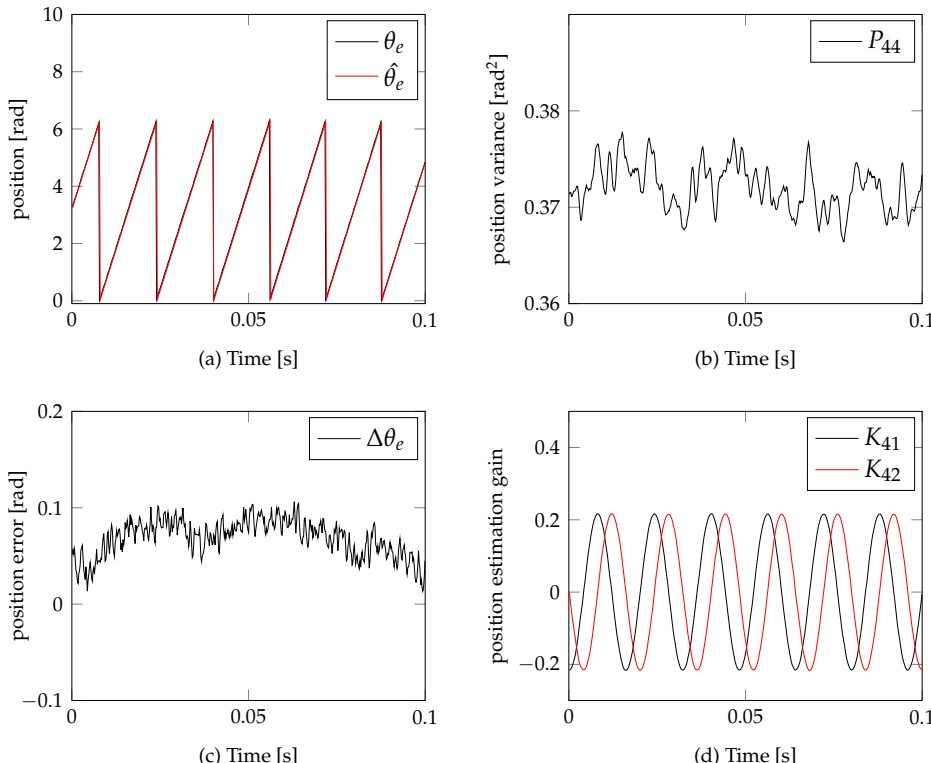

**Figure 4.** Simulated rotor position estimation performance at $f_b = 5$ kHz $f_s = 5$ kHz and $w_e = 400$ rads. (**a**) real and estimated rotor positions, (**b**) estimated position error, (**c**) estimated position variance, (**d**) position estimation Kalman gains.

Further simulation has shown, that slowing down the Kalman gain and covariance matrices calculation frequency could cause a rise in the rotor position estimation error, causing a loss of the synchronization and causing the rotor speed to drop to zero. The relation between the minimum Kalman gain and covariance matrices calculation frequency $f_{bmin}$ and the fundamental rotor electrical rotor frequency ($f_e = w_e/2\pi$) was investigated. To found the relation $\frac{f_{bmin}}{f_e}$ the simulations were run by decreasing frequency $f_b$ for fixed rotor speed reference set-point $w_{eref}$ until control was lost. The loss of control was easily determined from rotor speed not able to reach reference speed in simulation. For various rotor reference speeds $w_{eref}$ the minimum Kalman gain update frequency $f_{bmin}$ are shown in Figure 6a. From Figure 6a we can conclude that the higher the reference speed was set the higher Kalman gain update frequency was required. From Figure 6a we can see linear trend of $f_{bmin}$ to $f_e$. The steps in Figure 6a mainly are due to simulation strategy when frequencie $f_b$ were selected in steps by dividing switching frequency by integer value ($f_b = f_s / $ n, n = 1,2...). The Figure 6b is obtained from figure Figure 6a by dividing each $f_{bmin}$ by $f_e$. While the data in Figure 6b are the same we can see how many times we have to compute Kalman gain per rotor electrical frequency without lost control.

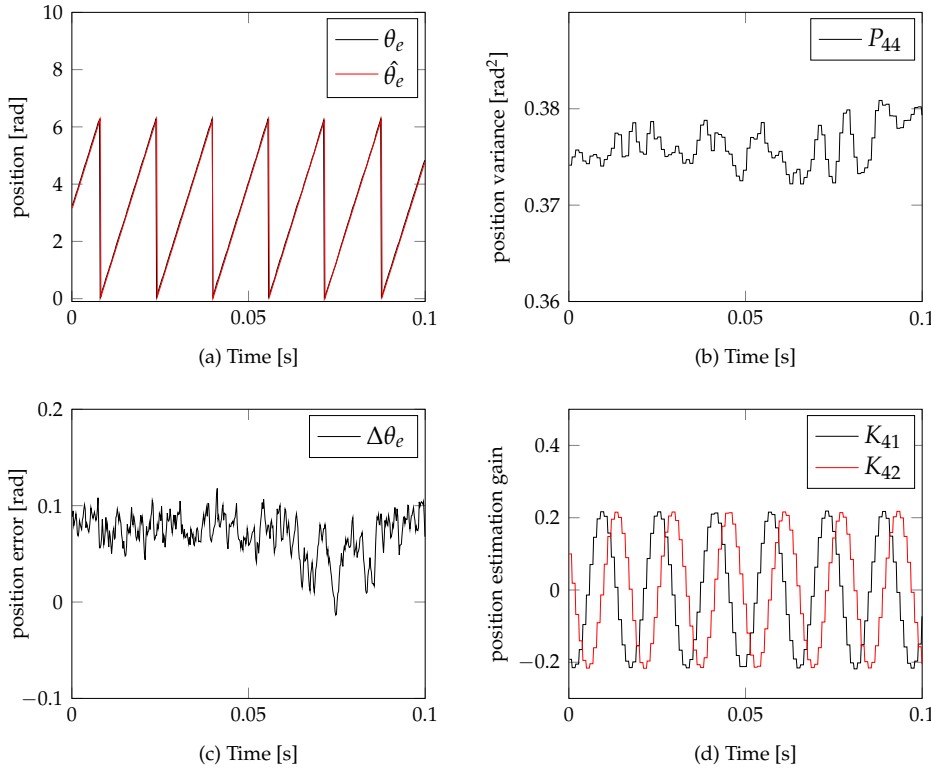

**Figure 5.** Simulated rotor position performance at $f_b = 1$ kHz $f_s = 5$ kHz and $w_e = 400$ rads. (**a**) real and estimated rotor positions, (**b**) estimated position error, (**c**) estimated position variance, (**d**) position estimation Kalman gains.

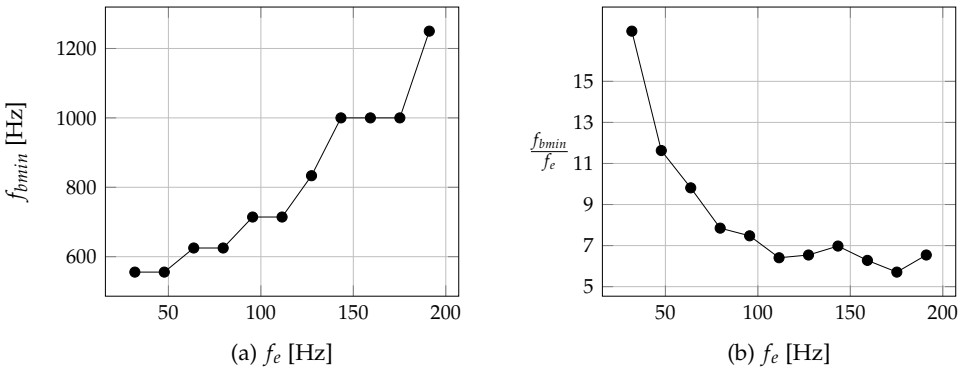

**Figure 6.** The relation between the minimum required Kalman matrices calculation frequency $f_{bmin}$ and the rotor electrical frequency $f_e$. (**a**) the relation between minimum required $f_{bmin}$ and $f_e$, (**b**) the relation between the ratio of $f_{bmin}$ to $f_e$ and the rotor electrical frequency $f_e$.

## 5. Implementation

For the implementation of the EKF algorithm in a real experimental drive system an NXP LPC1549 microcontroller was used. The LPC1549 microcontroller is a 32 bit ARM Cortex-M3 based microcontroller operating at a frequency of up to 72 MHz. It is a microcontroller characterized by a low cost and very low power consumption. It includes two 2 Msamples ADCs, four voltage comparators and a PWM/timer subsystem with four configurable multi-purpose state configurable timers. As the LPC1549 microcontroller does not support of floating point operations, all calculation was performed on integer type variables. Data type of the variables used in the implementation is 16 bit fixed point variable with 32 bit long words. For the implementation in C language and code compilation the

IAR compiler was used. Different optimization levels for speed comparison are given in Table 2.

**Table 2.** Comparison of the EKF steps execution time.

| Variable | Defining Equation | No Optimization [μs] | Compiler [μs] | Compiler + User [μs] |
|---|---|---|---|---|
| $\mathbf{P}_k^-$ | $\mathbf{P}_{k-1}\mathbf{F}_{k-1}\mathbf{P}_{k-1}^T + \mathbf{Q}$ | 97.2 | 56.8 | 13.4 |
| $\mathbf{K}_k$ | $\mathbf{P}_k^-\mathbf{H}^T(\mathbf{H}\mathbf{P}_k^-\mathbf{H}^T + \mathbf{R})^{-1}$ | 56.0 | 34.0 | 5.4 |
| $\mathbf{P}_k$ | $(\mathbf{I} - \mathbf{K}_k\mathbf{H})\mathbf{P}_k^-$ | 78.0 | 48.8 | 9.6 |
| $\mathbf{F}, \mathbf{B}$ | $\frac{\partial \mathbf{f}(\mathbf{x})}{\partial \mathbf{x}}$ | 4.4 | 3.2 | 3.2 |
| $\hat{\mathbf{x}}_k^-$ | $\hat{\mathbf{x}}_{k-1} + T\mathbf{f}(\hat{\mathbf{x}}_{k-1}) + \mathbf{B}\mathbf{u}_{k-1}$ | 7.2 | 4.4 | 4.4 |
| $\hat{\mathbf{x}}_k$ | $\hat{\mathbf{x}}_k^- + \mathbf{K}_k(\mathbf{y}_k - \mathbf{H}\hat{\mathbf{x}}_k^-)$ | 17.6 | 12.6 | 1.7 |
| *Total* | | 260.4 | 159.8 | 37.7 |

The EKF algorithm implementation using common matrices calculation procedures, with the IAR compiler and with no optimization, gave the longest total execution time 260.4 μs. Better results were obtained by setting the compiler optimization to a high level. However, the total execution time is still too long for a practical application. A good execution optimization could be achieved by discarding the common matrices calculation procedures, replacing them with simple arithmetic expressions and accounting for all zero elements and symmetry. In this case the total execution time drops to 37.7 μs and could be used for some practical applications.

However, in situations where very high switching frequency is desired, or other heavy calculations need to be performed, the execution time could still be too long. By implementing the EKF algorithm given in Figure 2 the total execution time could be reduced further. In experimental implementation the switching frequency is selected as 5 kHz and the background procedure call frequency is 1 kHz. In this case the total microcontroller usage for the EKF algorithm calculation is 6.21%. However, if the entire EKF calculation is performed in one switching period, the microcontroller uses as much as is 18.85% of its computational power. With these settings the total microcontroller processing time for EKF is reduced more than 3 times.

## 6. Experimental Setup and Results

The experimental test platform is shown in Figure 7. The test setup mainly consists of: a 30 W PMSM motor driven by an NXP inverter FRDM-MC-LVPMSM; other PMSM as a load; the NXP board Xpresso-LPC1549; DC voltage power supply 24V; 360P photoelectric incremental rotary encoder; personal computer (PC).

The PMSM parameters used in experimental setup are defined in Table 3. All measurements are done in real time by an LPC1549 microcontroller and data are transferred into a personal computer. The phase current was sampled by sensors while the phase voltages taken from control algorithm. The rotor position estimated by the EKF was compared to the signal obtained from a photoelectric incremental rotary encoder. In Figure 8a the estimated rotor position is compared with rotor position measured by the encoder, for the reference speed set to 400 rad/s. Figure 8b shows position variance and Figure 8d presents the Kalman gains for position measurement corrections. The switching frequency $f_s$ was set for 5 kHz, while the Kalman gain and all covariance matrices are calculated at the same frequency of $f_b = 5$ kHz.

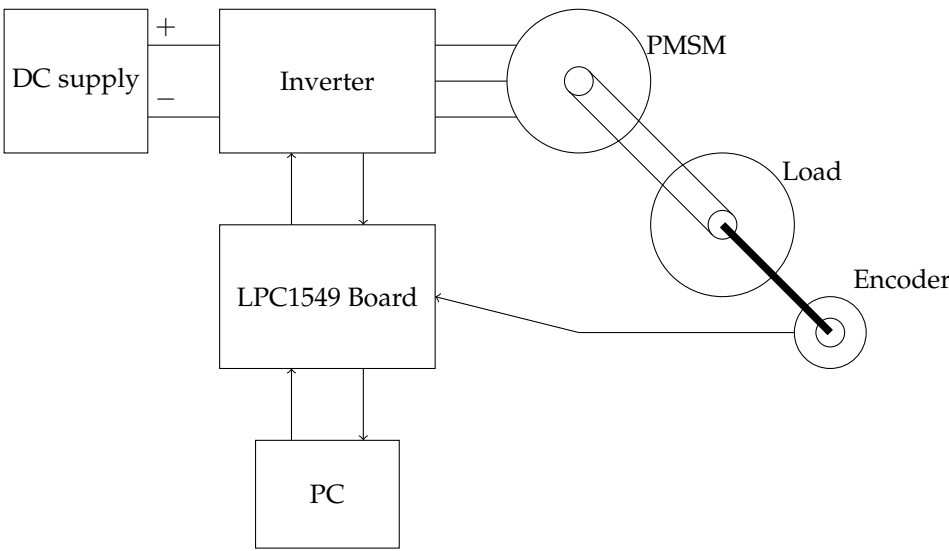

**Figure 7.** Experimental system configuration.

**Table 3.** PMSM parameters.

| Parameter | Symbol | Value |
|:---:|:---:|:---:|
| Stator resistance | $Rs$ | 1.2 Ω |
| Synchronous inductance | $Ld$ | 0.5 mH |
| Synchronous inductance | $Lq$ | 0.5 mH |
| Flux linkage | $\lambda_m$ | 0.007 Wb |
| Number of poles | $P$ | 8 |
| Nominal voltage | $V_n$ | 24 V |
| Nominal torque | $T_n$ | 0.063 Nm |
| Rated speed | $\omega_n$ | 4000 RPM |
| Rated power | $P_n$ | 30 W |

In Figure 9 the same measurements are performed, but with a reduced Kalman gain and covariance matrices calculation frequency $f_b$ of 1 kHz. To get results in Figure 9 the Kalman gain and covariance matrices are computed 5 times slower. Comparing the results presented in Figures 8b with 9b one can see, that the position error is about the same average value. More details about position error dependency are presented below. The position variance $P_{44}$ in Figures 8a and 9a is about the same average value [0.35 − 0.37]. This means that the variance of estimated position did not change with reduced Kalman gain and covariance matrices calculation frequency. The Kalman gains for position estimation in Figures 8d and 9d have the same amplitude peak values and have similar shapes, but the shape in Figure 9d are more discrete. A bigger discretization steps are mainly due to reduced calculation frequency of Kalman gain. During a short time ($1/f_b$) the Kalman gain matrix is assumed to be a constant value. And the Kalman gain matrix is used multiple times in control procedure until background procedure calculates new one.

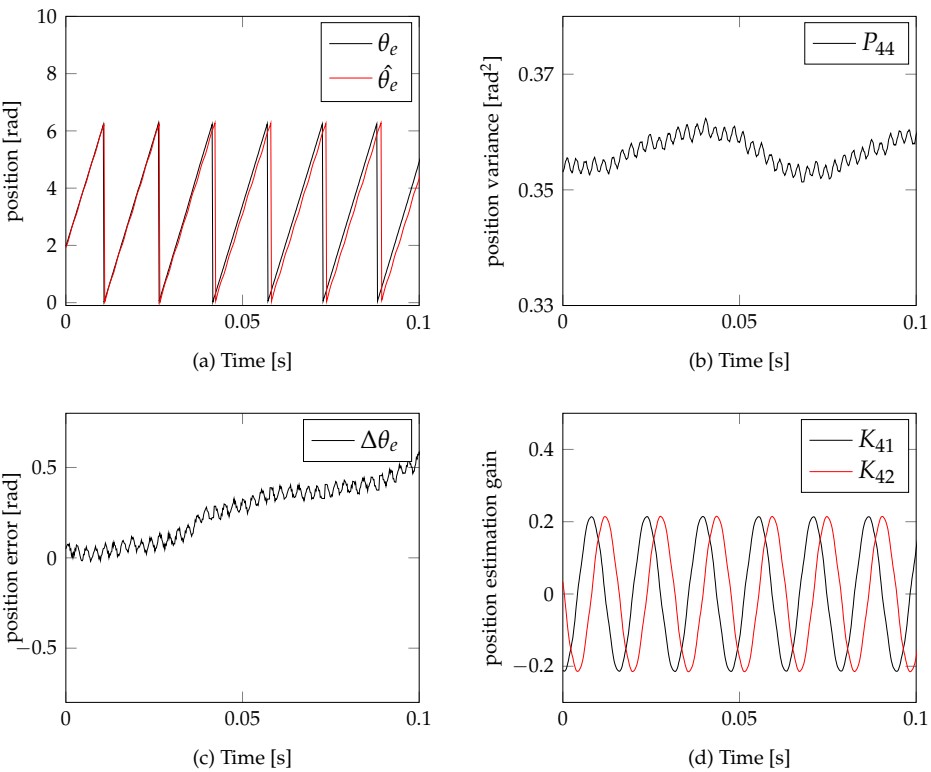

**Figure 8.** Experimental results of the rotor position estimation performance at $f_b = 5$ kHz $f_s = 5$ kHz and $w_e = 400$ rads. (**a**) measured and estimated rotor positions, (**b**) estimated position error, (**c**) estimated position variance, (**d**) position estimation Kalman gains.

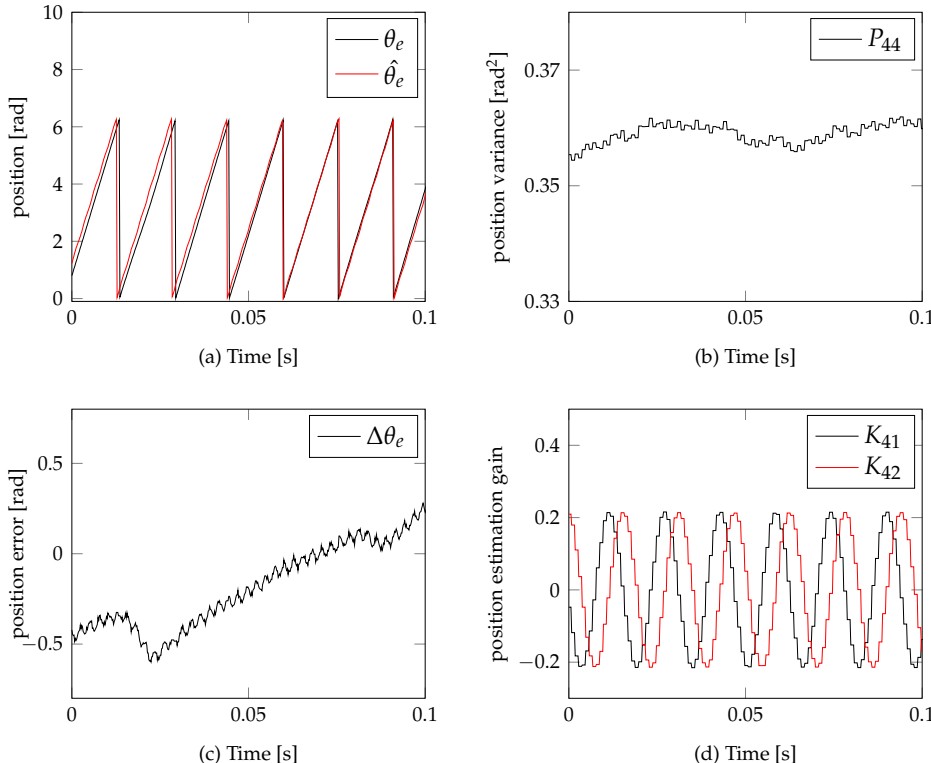

**Figure 9.** Experimental results ot the rotor position estimation performance at $f_b = 1$ kHz, $f_s = 5$ kHz and $w_e = 400$ rads. (**a**) measured and estimated rotor positions, (**b**) estimated position error, (**c**) estimated position variance, (**d**) position estimation Kalman gains.

Further experimental tests have shown that the position error did not increase when reducing the Kalman gain, and it's covariance matrices calculation rate, until some minimum frequency $f_{bmin}$ is reached. The position errors at various $f_b$ frequencies are shown in Figure 10. In Figure 10e the Kalman gain, and it's covariance matrices calculation update rate is 12 times slower ($f_b = f_s/12$) than the switching frequency, while the position error order is about the same, as when calculated in every switching instance.

However, slowing the Kalman gain and covariance matrices calculation update rate will cause the synchronization to be lost. By setting: $f_b = f_s/13$ and reference speed $w_{eref} = 400$ Hz the synchronization is lost, and the rotor speed drops to zero and reverse spin occurs as shown in Figure 11.

The last experimental tests have shown that the minimum Kalman gain and covariance matrices calculation update rate depends on the fundamental electrical rotor speed and this relation is very similar to simulation results.

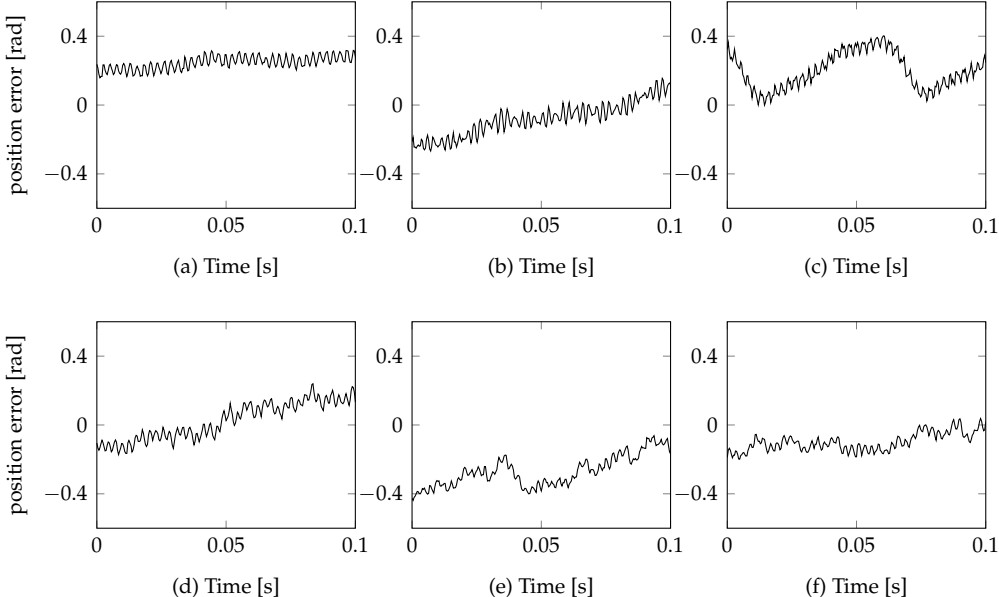

**Figure 10.** Estimated position errors at various Kalman matrices calculation update frequencies $f_b$. The switching frequency 5 kHz and the rotor electrical angular speed 400 rad/s. (**a**) $f_b = 5$ kHz, (**b**) $f_b = 2.5$ kHz, (**c**) $f_b = 1.25$ kHz, (**d**) $f_b = 500$ Hz, (**e**) $f_b = 454$ Hz, (**f**) $f_b = 416$ Hz.

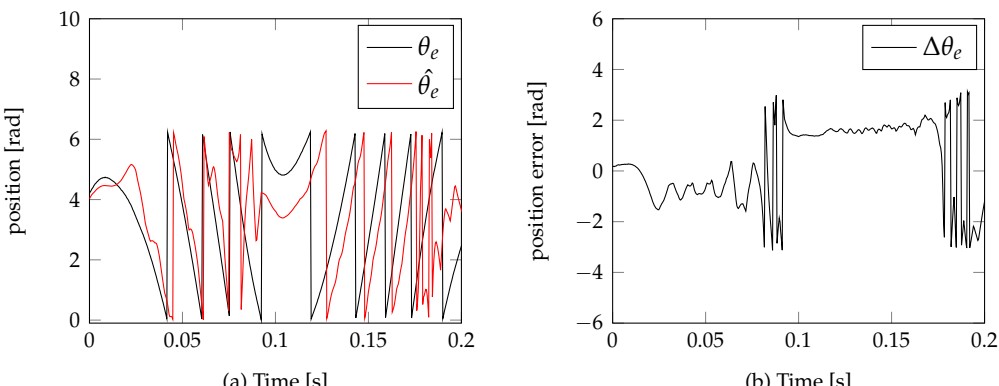

**Figure 11.** Synchronization lost at: $f_b = 384$ Hz, $f_s = 5$ kHz and reference speed $w_{eref} = 400$ rad/s. (**a**) measured ($\theta_e$) and estimated ($\hat{\theta}_e$) rotor positions, (**b**) estimated position error ($\Delta\theta_e$).

The Figure 12a depicts the relation between the minimum matrices calculation frequency $f_{bmin}$ and fundamental rotor frequency $f_e$. To obtain the relation of $\frac{f_b}{f_e}$ the experiments with different speed reference points was analyzed. For each speed reference point,

the minimum frequency $f_{bmin}$ was searched. The frequency $f_{bmin}$ considered enough high if the motor speed with constant reference speed was controllable. In the experiment, the frequency $f_{bmin}$ was reduced until the control was lost. The loss of the control was easy to see from the rotor speed not able to follow the speed reference set-point, the speed of the rotor dropped to zero. The synchronization was obviously lost. From this experiments it could be seen that the higher the reference speed, the higher update matrices calculation rate was needed. From Figure 12a the relation of minimum required $f_{bmin}$ and $f_e$ is linear. The Figure 12b presents the same measured data as in Figure 12a, by taking ratio $\frac{f_b}{f_e}$. From Figure 12b can be seen how many times per rotor electrical period, the Kalman gain and covariance matrices has to be computed without losing control. From Figure 12b we can conclude, that for high frequencies ($f_e > 50$ Hz) the minimum 7 times Kalman gain and covariance matrices have to be computed per one rotor electrical period. The lower ratio of the Kalman gain and covariance matrices calculation ratio ($\frac{f_b}{f_e} < 6$) will mostly cause lost control. The higher ratio ($\frac{f_b}{f_e} > 7$) of course will be good, but the higher ratio means more computation power are needed. Also, from Figure 12b we can conclude, that at low frequencies ($f_e < 50$ Hz) the ratio $\frac{f_b}{f_e}$ is higher than 7 is needed, but from Figure 12a we can see that the required minimum frequency $f_{bmin}$ is decreasing even if the ratio increase $\frac{f_{bmin}}{f_e}$.

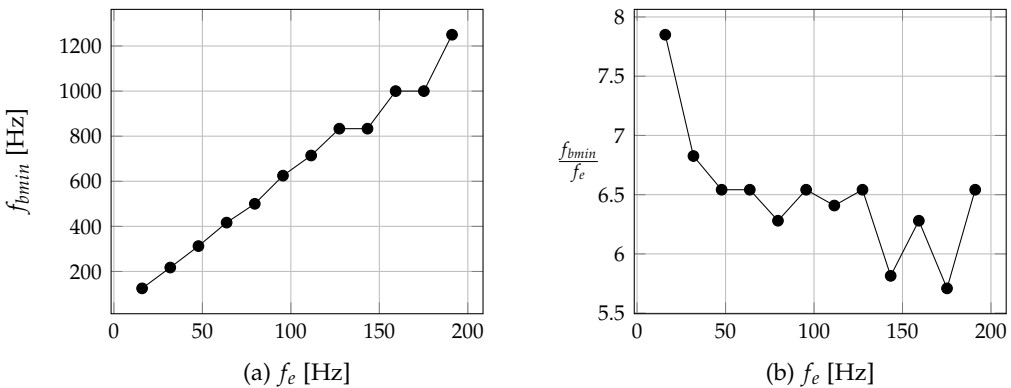

(a) $f_e$ [Hz]  (b) $f_e$ [Hz]

**Figure 12.** Experimentally measured relation between the minimum required Kalman matrices calculation frequency $f_{bmin}$ and the rotor electrical frequency $f_e$. (**a**) the relation between minimum required $f_{bmin}$ and $f_e$, (**b**) the relation between the ratio of $f_{bmin}$ to $f_e$ and the rotor electrical frequency $f_e$.

ARM Cortex-M3 LPC1549 processor usages for executing EKF estimator with various background frequencies are given in Table 4. The data from Table 2 was used for calculating usages. Also, by applying the role that at least 7 times per one rotor electrical period Kalman gain and covariance matrices have to be computed, we get maximum allowed electrical frequency ($f_{emax}$) in control. That level of optimization to choose is a trade-off decision between processor usage and the maximum allowed electrical frequency in the system.

**Table 4.** Processor usages for executing EKF estimator with different background frequencies, $f_s = 5$ kHz.

| $f_b$ | CPU Usage | $f_{emax}$ |
|---|---|---|
| 5 kHz | 18.85% | 714 Hz |
| 2.5 kHz | 10.95% | 357 Hz |
| 1.25 kHz | 7.0% | 179 Hz |
| 500 Hz | 6.21% | 71 Hz |

From Table 4 can be seen that even reduction of background frequency by two times from 5 kHz to 2.5 kHz gives a good time optimization. The further reduction gives less win of the processor time for each step.

## 7. Conclusions

This paper proposes optimized EKF estimation method for PMSM sensorless control with low execution time. The computational methods used to simplify the EKF estimator and their implementation in fixed-point arithmetic are discussed. Experiments and simulation have been carried out to evaluate the performance and the computing cost of the EKF based on Kalman gain and all covariance matrices calculation in separation from the prediction and update steps. The primary conclusions are summarized as follows:

1.  Separation of the Kalman gain and all covariance matrices calculation from prediction and update steps could provide the same accuracy with less execution time of the processor.
2.  The minimum required Kalman gain and all covariance matrices update ratio depend on the fundamental electrical frequency, the higher electrical frequency the higher update frequency required.
3.  The estimated rotor position error did not increase until minimum required Kalman gain and all covariance matrices update ratio is reached.

**Author Contributions:** Conceptualization, J.D; methodology, J.D.; software, J.D.; validation, J.D., V.S. and K.Ł.; formal analysis, J.D, V.S. and K.Ł.; investigation, J.D.; resources, J.D.; data curation, J.D.; writing-original draft preparation, J.D.; writing-review and editing, J.D., V.S. and K.Ł.; visualization, J.D.; supervision, J.D.; All authors have read and agreed to the published version of the manuscript.

**Funding:** This research received no external funding.

**Conflicts of Interest:** The authors declare no conflict of interest.

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
