# Peer review of "Implementation of Extended Kalman Filter with Optimized Execution Time for Sensorless Control of a PMSM Using ARM Cortex-M3 Microcontroller"

_energies, doi:10.3390/en14123491_

Round 1

Reviewer 1 Report

In this study, the authors presented a  Extended Kalman Filter for Sensorless Control of a PMSM.

Some concerns are listed as follows:

The abstract must be improved (present all the novelties).

The motivation and the motivation should be more explained in the introduction.

Indeed, more detail about what the paper is about with more bibliographic references should be inserted.

The proposed results and methodology should be explained in more detail, and some remarks should be added.

 In the simulation section: Discussion, figures... should be improved.

Some comparative results using other recently published  Kalman Filter   structures should be provided to verify the merits of the proposed control scheme.

In the conclusions section, new features of the proposed work should be further highlighted. In addition, some future work must be mention.

Author Response

Hello,

Reviewer 2 Report

The authors propose a strategy to decouple the EKF matrix computation from the prediction and measurement update phases to minimize the overall time consumption of the EKF algorithm. The strategy was simulated in the Matlab programming language and implemented on an ARM Cortex-M3 microcontroller.

It looks like an interesting study, but there are some areas for correction.

1. There are insufficient references in this study. Please write your introduction more faithfully based on existing studies. Please cite the latest research paper as a reference and strongly emphasize the necessity of this study.

2. I think the explanation of Figures 7,8,9,10 is very insufficient. Please do not simply describe the characteristics in the picture, but write the meaning of each characteristic in more detail.
There is no explanation for Figure 10.b. In particular, it is necessary to explain the part where the value of fb/fe is not linear and has pulsation depending on the magnitude of the frequency.

3. Since this study has not been validated with the experimentally proposed method, the authors should supplement with more detailed descriptions.
Or, if you configure the experimental environment and add the test results, I am sure that it will be a strong basis to support the ideas proposed by the authors.

Author Response

Hello,

Reviewer 3 Report

This paper presents EKF-based sensorless control of a PMSM. The paper is generally well written with sufficient analysis. It can be considered for acceptance if the authors can address the following issues.

  1. Please ensure that all the variables have been defined or explained clearly.
  2. The EKF has been widely used for state estimation, e.g., Journal of Power Sources 332 (2016): 389-398; Applied energy 204 (2017): 1264-1274. Such works can be mentioned to widen the literature review and relevant explanation.
  3. The performance of EKF is largely linked to the parameter tunning, and the associated tunning optimization method has been studied in the literature, e.g., DOI: 10.1109/TTE.2020.3032737. Such works are very relevant to the presented method.
  4. The reviewer suggests to add more details about the experiments.
  5. Some error spike can be observed from the estimation results in figure 9. Please comment on this.
  6. The conclusions are not well written. Please give primary findings with statistical results.

Author Response

Hello,

Round 2

Reviewer 2 Report

Thank you for your reply.

Reviewer 3 Report

This paper can be accepted.